# Renewable Energy in Indonesia: Current Status, Potential, and Future Development

**Nugroho Agung Pambudi** [1,2,*], **Ridho Alfan Firdaus** [1], **Reza Rizkiana** [1], **Desita Kamila Ulfa** [1], **Muntasy Syahrul Salsabila** [1], **Suharno** [2] **and Sukatiman** [3]

1. Energy and Society Laboratory, Mechanical Engineering Education, Universitas Sebelas Maret, Surakarta 57126, Indonesia
2. Mechanical Engineering Education, Universitas Sebelas Maret, Surakarta 57126, Indonesia
3. Civil Engineering Education, Universitas Sebelas Maret, Surakarta 57126, Indonesia
* Correspondence: agung.pambudi@staff.uns.ac.id

**Abstract:** The current use of fossil fuels has a significant impact on increasing greenhouse gas (GHG) emissions. Subsequently, renewable energy is significantly needed to reduce GHG, thereby limiting the impact of extreme weather and climate while ensuring reliable, timely, and cost-effective supply. As a big country with a huge amount natural resource, the demand for renewable energy in Indonesia has increased along with the rise in consumption. Following this, energy consumption increased by 0.99%, which was approximately 939.100 million BOE in 2021 for biogas, oil, electricity, natural gas, coal, LPG, biodiesel, and biomass. Energy consumption in several sectors including transportation has the largest energy consumption with approximately 45.76% of oil. In industries and households sector, the consumption rates are 31.11% for boiler steam generation purposes and 16.89% for electricity as well as LPG. Furthermore, the commercial sector consumes 4.97% of energy for lighting and air conditioning, while the remaining 1.27% is used for other sectors. Meanwhile, Indonesia has high potential for renewable energy at 419 GW including 75 GW of hydro energy, 23.7 GW of geothermal, 32.6 GW of bioenergy, 207.8 GW of solar, 60.6 GW of wind, and 19.3 GW of micro-hydro. Therefore, the main focus of this paper is to provide a detailed analysis of the current status, prospects, and information on Indonesia's renewable and sustainable energy sources. Furthermore, the novelty of this research entails updating the latest data related to renewable energy and its availability in Indonesia. The essence is to portray a picture of its potential development in the future.

**Keywords:** carbon emission; energy consumption; potential energy; renewable energy

## 1. Introduction

Energy production and consumption are essential for economic growth in all countries [1]. Therefore, the current use of fossil fuel, such as coal, oil, and natural gas, has a significant impact on carbon emissions. This causes an increase in greenhouse gas emissions (GHG), leading to an unstable climate, as well as a rise in the earth's temperature and sea level [2–4]. The research proved that $CO_2$ emissions had made the biggest contribution to climate change [5–7]. However, fossil energy cannot be renewed, hence its availability decreases. Future growth in the energy sector will likely lead to a shift toward renewable energy, which can help reduce GHG emissions by limiting the impact of extreme weather and climate while ensuring reliable, timely, and cost-effective supply [8,9].

According to Yin et al. (2022), climate change reduces carbon emissions, and this depicts the existence of dynamic factors. The impact of forest carbon sequestration efficiency was analyzed using the data envelopment method. In addition, gross domestic product (GDP) per capita urbanization and the highway network were also evaluated, and it was reported that these factors have a significant positive impact on carbon sequestration

efficiency. The entire import and export activities tend to have a significant negative impact. The results obtained are extremely important for improving financial investment efficiency, forest quality, and carbon sinks [10].

Efforts to reduce $CO_2$ emissions can be realized through poverty alleviation initiatives, which is highly prioritized in developing countries. This is also in a bid to achieve Sustainable Development Goal 7 (SDG7), namely affordable and clean energy, as well as reduced $CO_2$ emissions. In addition, these efforts also has an impact on state policies in terms of realizing a sustainable environment through renewable energy, economic activities, and trade freedom [11].

In 2020, Indonesia succeeded in reducing its GHG by 25.93%, but in 2021, it weakened to 23.55% [12]. Therefore, low-carbon development actions should be carried out optimally for 2022 and subsequent years by increasing government programs and budgets [13]. The actions can be reforestation, prevention of deforestation, increasing renewable energy capacity, and energy efficiency. Therefore, restoring economic and social activities particularly after the COVID-19 pandemic needs to be in line with efforts to reduce GHG emissions [14]. The vision of switching energy sources from fossil to renewable can be called transition [15]. Several countries have strengthened international cooperation to facilitate access to clean, renewable and efficient energy technologies, including Indonesia [16,17].

According to data from the government [18], the country total primary energy supply without biomass in 2020 was 201.6 million TOE, as shown in Table 1. Figure 1 indicates that Indonesia is still dependent on fossil, despite its transition to the use of renewable energy. This transition is realized by increasing the percentage of the renewable energy mix from 11% in 2021 to 23% in 2025 and 31% in 2050 [19–21]. The percentage of the fossil energy mix is projected to decrease despite the increase in the demand for primary fossil energy supply [22].

**Table 1.** Primary energy supply in Indonesia in 2020 [18].

| Fuel Type | Primary Energy Supply (TOE) | Total (%) |
|---|---|---|
| Coal | 77.5 million | 38.5% |
| Oil | 66.2 million | 32.8% |
| Gases | 35.2 million | 17.4% |
| Renewable energy including hydro, geothermal, solar, wind, biofuel, and biogases | 22.7 million | 11.3% |

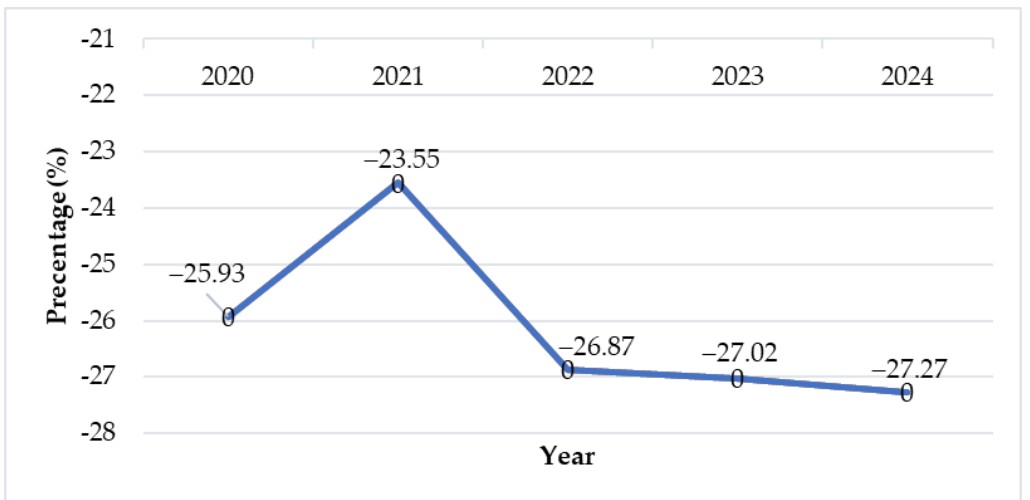

**Figure 1.** Projected GHG emissions reduction [12].

The demand and potential for renewable energy are increasing largely due to the significant growth in energy consumption globally. However, Indonesia is a country that has a large enough potential for renewable energy, hence it should be optimized in future [23,24]. The novelty of this research entails updating the latest data related to renewable energy and its availability in Indonesia. The essence is to portray a picture of its potential development in the future.

This research is mainly focused on providing a detailed analysis of the current status, prospects, and information on Indonesia's renewable and sustainable energy sources. Based on these objectives, the results obtained are expected to serve as a reference in future studies and boost one's knowledge about the potentials of renewable energy. It can also be used to enact energy-related policies, especially the renewable type, intended for future purposes. Figure 2 shows a layout of the content including methods (Section 2), energy policy in Indonesia (Section 3), energy consumption in Indonesia (Section 4), available energy sources (Section 5), current energy situation (Section 6), potential and future development (Section 7), and conclusion (Section 8).

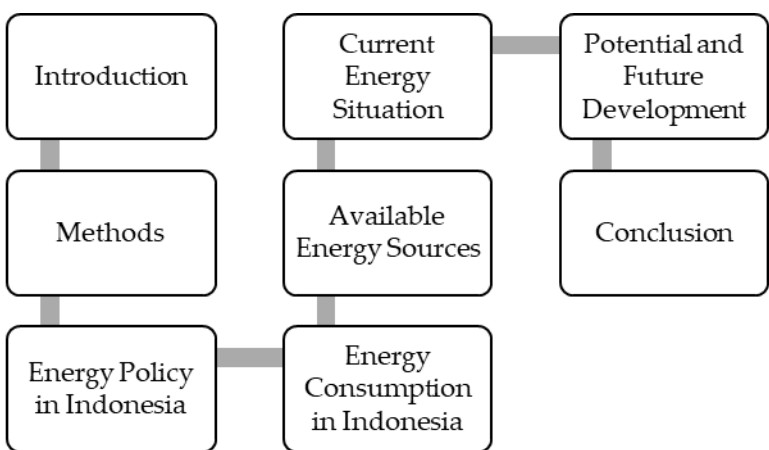

**Figure 2.** Layout of the content.

## 2. Methods

As shown in Figure 3, this systematic review is carried out based on the following steps.

(1) Identification of the topic;
(2) Literature study;
(3) Screening;
(4) Analysis and synthesis;
(5) Compilation of article review.

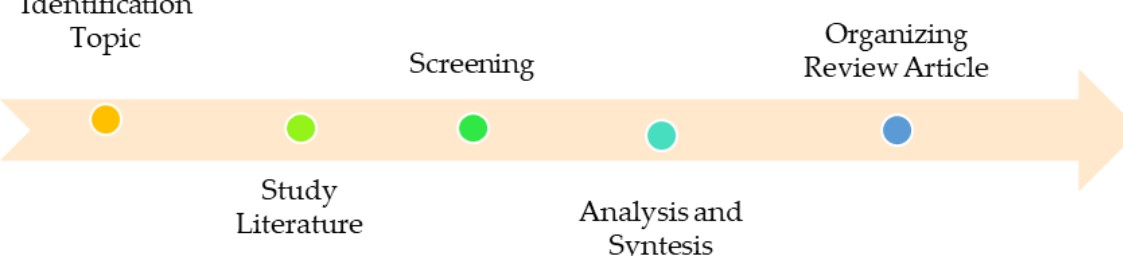

**Figure 3.** Systematic review of the entire process.

Referring to the research carried out by Ramdhani, et al. (2014), the following steps were employed in conducting an article review (1) selecting the topic to be reviewed, (2) tracking the appropriate or relevant articles, (3) analyzing and synthesizing diverse literatures, as well as (4) compiling writings for article review [25]. Furthermore, the

synthesis step is the most relevant one. This is performed by collecting various related articles and then compiling them into a conceptual or empirical analyses relevant to the research conducted.

## 3. Energy Policy in Indonesia

According to the Indonesian Ministry of Energy and Mineral Resources, the government had joined the Clean Energy Demand Initiative (CEDI). This is a way of supporting the international community in implementing climate change mitigation and improving the green economy. Therefore, the president's directives concerning the CEDI is needed to accelerate the steps necessary for achieving the nationally determined contribution (NDC) and net zero emissions (NZE), targeted at 2030 and 2060, respectively. Indonesia's transformation policy towards renewable energy should be promoted and strengthened. Furthermore, it has a vision and mission of achieving 23% renewable energy in the primary energy mix by 2025, with a relative decrease in emissions of from 29 to 41% [11,26].

Generally, most ASEAN countries allow the spread of only 16.9% renewable energy sources, with a 6.3% gap. To close this gap, each of these nations, including Indonesia, contributes to the rise in renewable energy share [27,28]. Moreover, the future national energy policy is enacted in government regulation no. 79 of 2014. This law serves as a guideline for the independent realization of national energy management and security for sustainable development [29].

## 4. Current Energy Situation

Energy usage in Indonesia is still significantly dependent on non-renewable sources, known as fossil fuels, which are still higher than renewable energy [30]. The country's energy balance in 2021 is shown in Table 2. In 2021, coal was the largest supplier at 558.782 million BOE and for renewable sources it is only 180.509 million BOE. Solar energy was the smallest supplier at 0.789 million BOE. Coal was also extensively used in the industrial sector with consumption of 87.820 million BOE, and dependent on fuel with a total consumption of 429.999 million BOE, part of which was satisfied from imports, and natural gas of 119.647 million BOE. Meanwhile, in 2021, only biogas energy was utilized in the household sector at a consumption rate of 0.180 million BOE.

**Table 2.** Indonesia energy balance in 2021 (million BOE) [31].

| Fuel Type | Primary Energy Supply | Power Plant | Final Energy Consumption | | | |
| --- | --- | --- | --- | --- | --- | --- |
| | | | Industry | Transportation | Household | Other |
| Coal | 558.782 | 470.962 | 87.820 | 0 | 0 | 0 |
| Oil | 133.009 | 17.512 | 25.776 | 388.157 | 2.657 | 10.788 |
| Natural gas | 324.608 | 83.804 | 88.841 | 0.066 | 0.308 | 0.701 |
| Hydro | 45.948 | 45.948 | 0 | 0 | 0 | 0 |
| Geothermal | 29.533 | 29.533 | 0 | 0 | 0 | 0 |
| Solar energy | 0.789 | 0.789 | 0 | 0 | 0 | 0 |
| Wind energy | 1.071 | 1.071 | 0 | 0 | 0 | 0 |
| Biofuel | 65.567 | 0 | 0 | 0 | 0 | 0 |
| Biogases | 0.180 | 0 | 0 | 0 | 0.180 | 0 |

In 2021, coal also was still the largest contributor to electricity generation, with a total of 470.962 million BOE. The energy used in electricity is only 114.762 million BOE for renewable energy, which is smaller to fossil. The largest renewable energy supplier is hydro with a total of 45.948 million BOE, while solar is the lowest with a total of 0.789 million BOE.

Energy is the most important part of survival in this world. Humans' dependence on fossil continuously impacts the occurrence of a crisis [32], as marked by the continued decline in oil and natural gas reserves yearly [18].

The data in Figure 4a shows a decrease in oil reserves by 6% and 8% in 2019 and 2020 from 2018. In Figure 4b, natural gas reserves also decreased by 0.53% and 5% in both years.

In contrast, renewable energy production increased compared to the previous year, mainly due to the rise in the production of hydro, geothermal, and solar power plants [18].

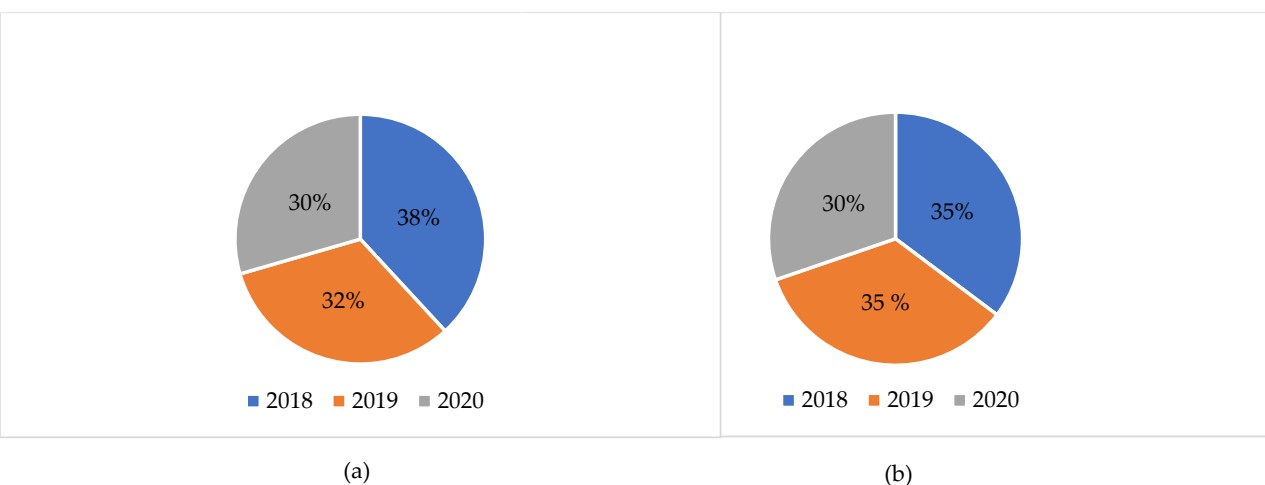

(a)                                                    (b)

**Figure 4.** (**a**) Oil reserve; (**b**) natural gas reserve [31].

The production of solar power plants (Figure 5a) increased from 4.56 GWh in 2018 to 5.66 GWh in 2021. Meanwhile, utilization of hydropower also increased from 10,729 GWh in 2018 to 11,869 GWh in 2021. Similarly, geothermal increased from 4013 GWh in 2018 to 4217 in 2021 (Figure 5b). This means that renewable energy is starting to increase and reduce the use of fossil sources [31].

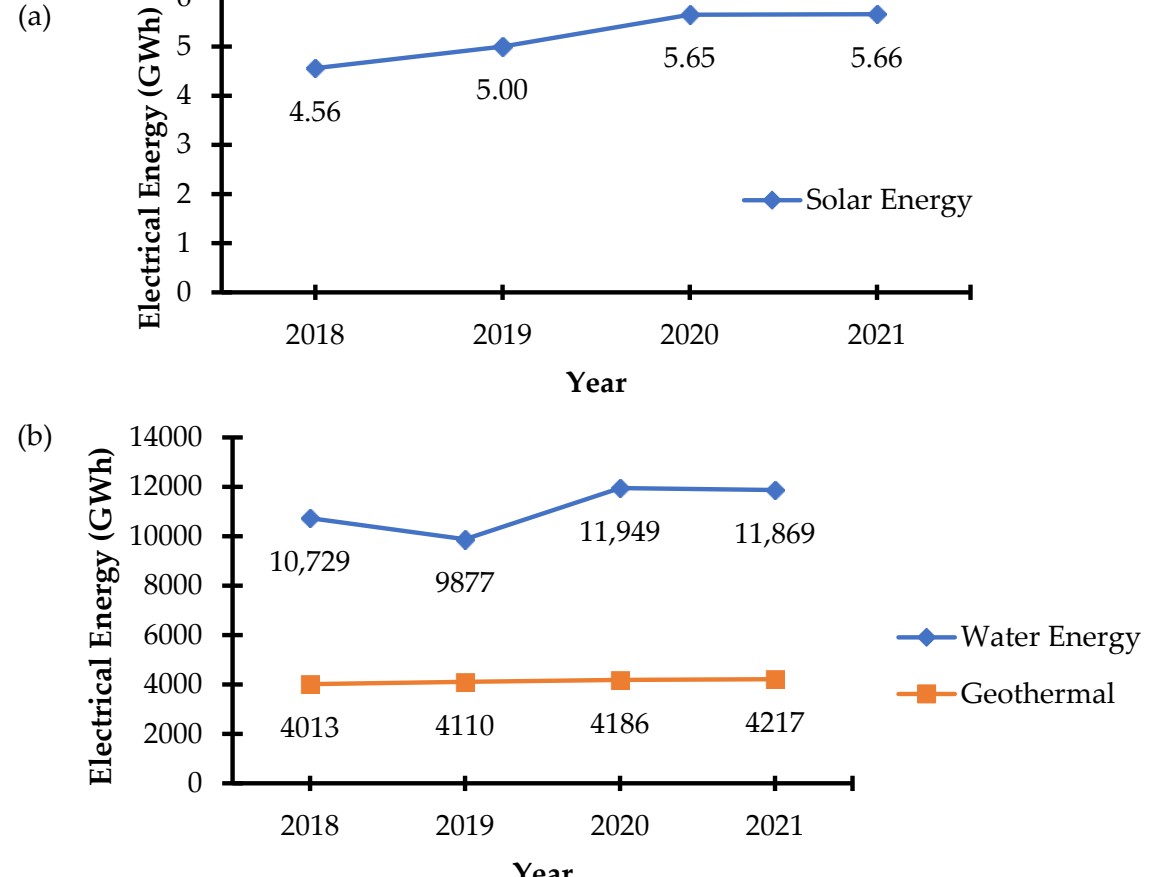

**Figure 5.** (**a**) Solar energy production; (**b**) production of hydro and geothermal power plant [31].

## 5. Available Energy Sources

Energy sources, such as fuel, electricity, mechanical and heat, are part of the basic needs of people in any country, including Indonesia [33,34]. It plays an important role in the development process of the social and environmental, which supports the national economy [35,36]. Furthermore, energy is also needed as the spearhead of various sectors of life such as technology, information, agriculture, education, health, and transportation [37,38]. Over time, the demand is increasing rapidly in line with economic and population growth [39–41]. Therefore, the availability of sustainable energy is important in maintaining sustainable development [42,43].

Indonesia has a wealth of natural resources, which can be used to produce energy directly or through a transformation process [44,45]. It consists of fossil primary sources such as oil, gas, and coal, as well as renewable ones such as hydro, geothermal, mini- and micro-hydro, solar, wind power, nuclear, and others [46,47]. Some of these sources can be processed to fulfil the community's needs, and the management should refer to the principles of sustainable development [47–49].

The energy balance in Indonesia has continuously changed over the years. Primary energy supply without biomass from 2015 to 2020 showed an increase from 169.8 million TOE to 201.6 million TOE, with an average growth of 3.5% per year. Meanwhile, the energy production in 2020 showed a total of 443.1 million TOE, with 94.9% from fossil including oil, gas, and coal [18].

Currently, the share of final energy consumption is still dominated by fossil fuels, which are found in almost all regions, including the islands of Sumatra, Java, and Kalimantan. The supply of oil reserves is approximately 4.17 billion barrels, with 2.44 billion barrels reserved. Meanwhile, natural gas reserves are 62.4 trillion cubic feet with proven reserves of 43.6 trillion cubic feet. Oil and gas reserves are estimated to be available for up to 9.5 years and 19.9 years, respectively. This is with the assumption that there are no new discoveries, and the level of oil production is approximately 700,000 barrels of oil per day (BOPD) and gas of 6 billion standard cubic feet per day (BSCFD) [50].

Figure 6 shows a decrease in oil and gas production due to a natural decline in reservoir performance and the inability to discover new large reserves [50]. Therefore, oil and gas reserves are predicted to decline continuously until 2024. The country projected that the remaining oil reserves in 2024 will be 1137.86 MMSTB, or a decrease of 48.56% from 2020 (Figure 7). This will also occur with natural gas, which will experience a decline of 22.02%.

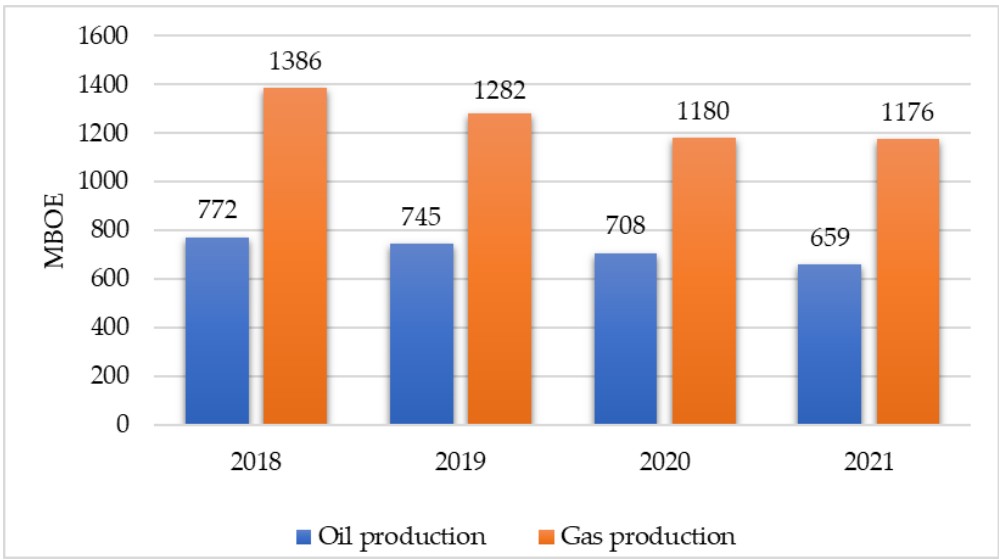

**Figure 6.** Oil and gas production 2018–2021 [50].

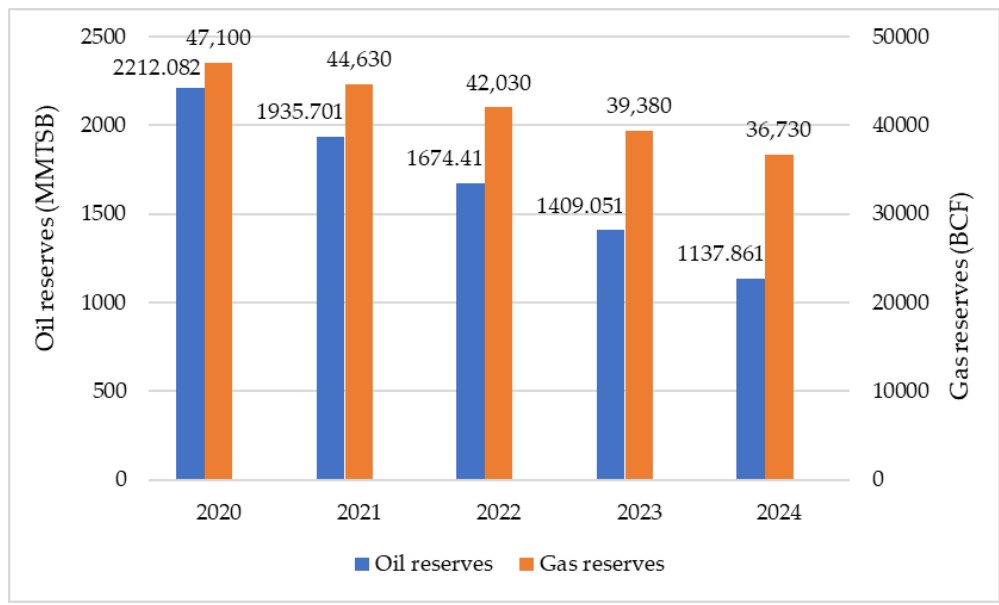

**Figure 7.** Indonesia's estimated oil and gas reserves in the years 2020–2040 [50].

The increasing demand for oil and gas energy is indicated by a rise in fuel oil production and imports in 2021 compared to 2020 (Figure 8). Its use is considered more economically profitable than other conventional energy, especially in the transportation sector as the largest energy user. Most of the transportation sector uses fuel oil because using renewable energy is considered not economical; hence, oil is still the best option.

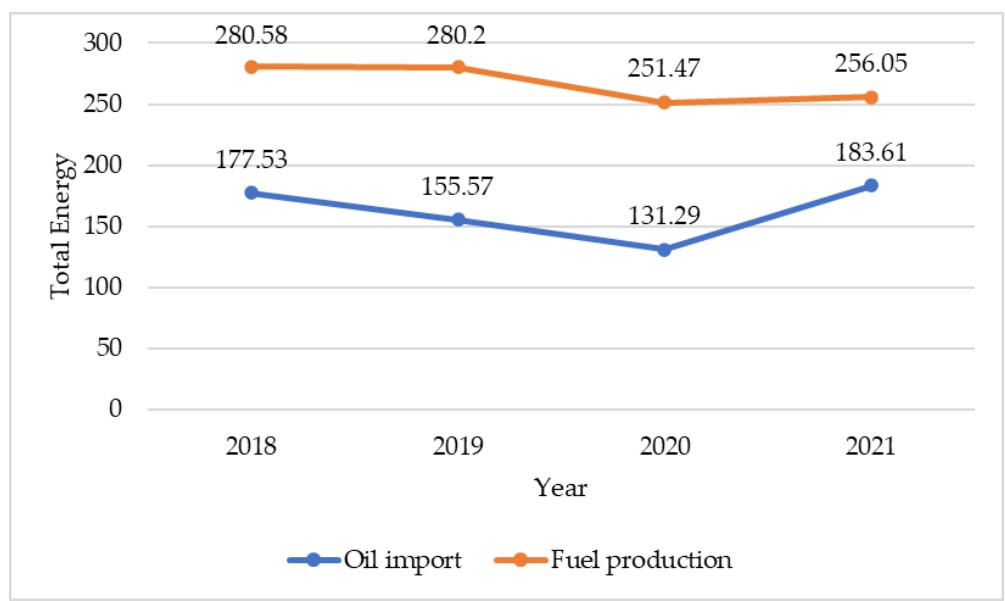

**Figure 8.** Annual domestic fuel production and fuel imports from 2018 to 2021 [50].

In addition to oil and gas, coal is a non-renewable natural resource with strategic value regionally and nationally. This natural resource is the mainstay of the Asia Pacific region in providing affordable and cheap sources, especially in the current situation of pandemic and the Russian–Ukrainian War. Coal reserves in Indonesia are spread across 21 provinces with 38.84 billion tons at an average production of 606.22 million tons produced per year. This is a 7.2% increment compared to 2020, which was 566 million tons, as shown in Figure 9. The coal reserves are estimated to be available for the next 65 years, assuming no new reserves are found. Furthermore, there are also coal resources of 143.7 billion tons, with the largest

located in Kalimantan with 62.1% or 88.31 billion tons of resources and 25.84 billion tons of reserves. Coal is also found in Sumatra, with a total of 55.08 billion tons of resources and 12.96 billion tons of reserves [50].

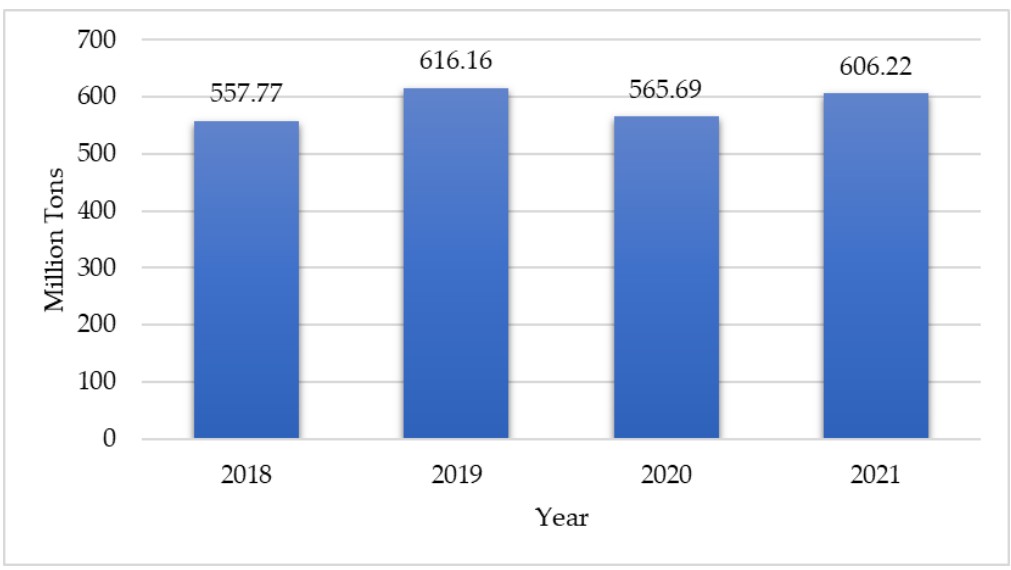

**Figure 9.** Annual coal production 2018–2022 [50].

The use of coal is divided into two, namely as raw material and fuel. Its use as raw material includes the manufacture of coal briquettes, metal processing, coal liquefaction, gasification, and upgrading. Meanwhile, it is used in the power generation sector, industry, small businesses, and households as fuel [50,51]. Coal is an essential resource of state revenue, which is economically very important. Therefore, its management needs to be carried out in an optimal, transparent, accountable, and fair manner to provide great benefits to the community [52–54]. Government policies in terms of supporting the development of coal mining should also pay attention to environmental changes, both nationally and internationally [55–57].

There is a need for renewable energy sources in Indonesia due to the high probability of a decrease in the availability of non-renewable energy sources. The enormous potential and use of renewable energy increased from 4.9% in 2015 to 11.3% in 2020 due to the rise in the share of biofuels and its use in the construction of off-grid power plants such as hydro, geothermal, solar power plants, etc. Presently, renewable energy supply in Indonesia is 22.7 million TOE or 11.3% consisting of hydro, geothermal, solar, wind, biofuel, and biogas. Meanwhile, the production is only about 5.1% of the national energy production [50].

## 6. Energy Consumption in Indonesia

Energy is globally needed in daily consumption and production activities in the industry, transportation, and agriculture sectors [58]. As a natural resource, it is used for the prosperity of the community, hence, proper management is needed to ensure sustainable development. The government has targeted a 17% reduction in final energy consumption by 2025 and a 1% decrease in energy intensity. Furthermore, approximately 10% to 30% reduction is targeted at the industrial, transportation, commercial, and household sectors [59]. Energy consumption is classified based on type and sector in the following section [31].

### 6.1. Energy Consumption Per Type

In Indonesia, energy consumption increased by 0.99% or 939.100 million BOE in 2021. This consumption included 45.72% biogas oil consisting of gasoil, biodiesel, and blended products in values of 15.76%, 7.10%, and 22.86%, respectively (Figure 10). Other energy

consumption includes oil, electricity, natural gas, coal, LPG, biodiesel, biogas, and biomass. The amount of energy per type is presented in detail in Table 3.

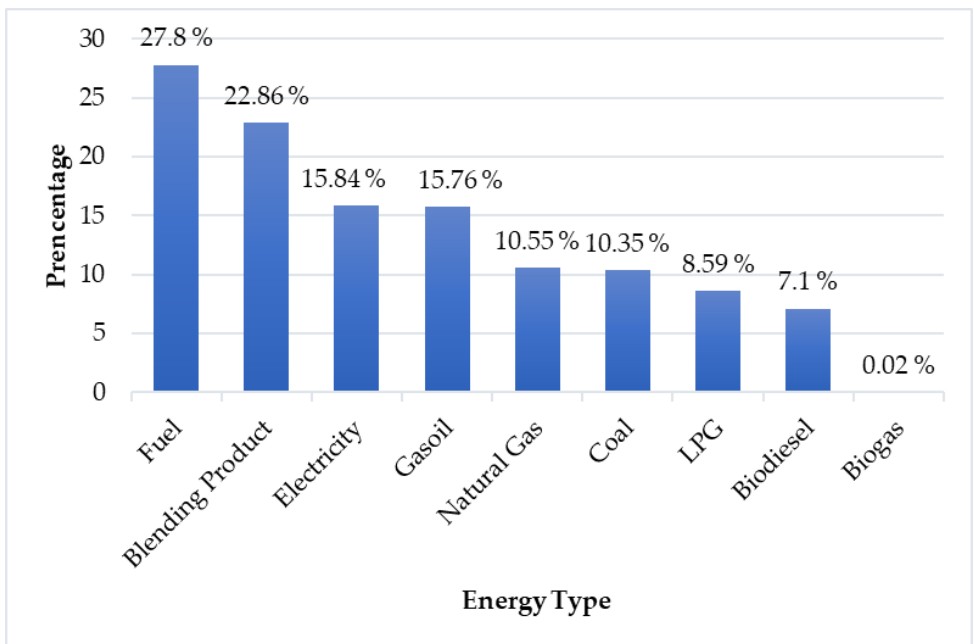

**Figure 10.** Energy consumption per type 2021 [31].

**Table 3.** Total energy per type (Million BOE) [31].

| No. | Energy Type | Total |
|---|---|---|
| 1. | Gasoil | 133.767 |
| 2. | Biodiesel | 60.292 |
| 3. | Blended product | 194.059 |
| 4. | Oil | 60.292 |
| 5. | Electricity | 194.059 |
| 6. | Natural gas | 89.557 |
| 7. | Coal | 87.820 |
| 8. | LPG | 72.921 |
| 9. | Biogas | 180 |
| 10. | Biomass | 60.392 |
| | Total | 939.100 |

Fossil energy is used as a temporary energy sources in the country, especially during the transition period before it is converted to 100% renewable energy in power plants. Natural gas was used as some form of fuel support for intermittent renewable energy plants, while minerals were mainly utilized for downstream processes. However, the government has started reducing the use of coal as an energy resource by adopting CCS/CCUS (carbon capture, utilization, and storage) technology, using dimethyl ether (DME) to replace LPG and increasing the added value of minerals through domestic downstream. In Indonesia, the energy sector emissions in 2021 amounted to 530 million tons of $CO_2e$. It was predicted that increased peak emissions to 706 million tons of $CO_2e$ are bound to occur around 2039. However, they will be significantly reduced after 2040, following the completion of fossil plant contracts [60].

*6.2. Energy Consumption Per Sector*

The energy consumption sector, including transportation, industry, household, commercial firms, etc., are shown in Figure 11. The largest energy consumption is in the

transportation sector, which consumes approximately 45.76% of gasoline. In industries and household, the consumption rates are 31.11% for boiler propulsion purposes and 16.89% for electricity and LPG consumption. Furthermore, the commercial sector consumes 4.97% of energy for lighting and air conditioning in elevators and escalators, while the remaining 1.27% is used for other sectors. The energy consumption per sector has a total of 909,244.973 million BOE, as shown in Table 4.

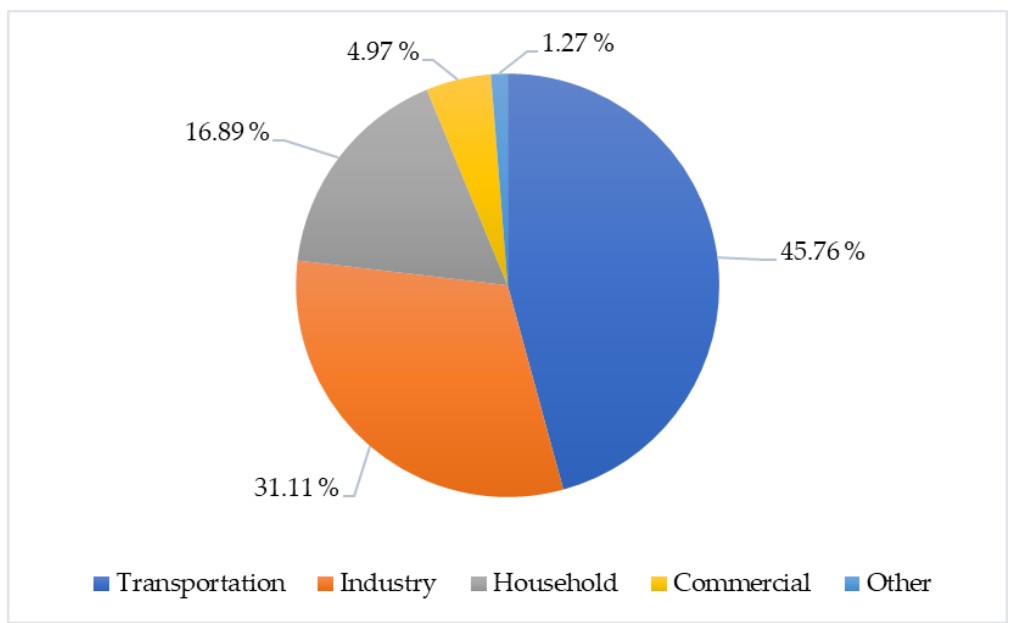

**Figure 11.** Energy consumption per sector [31].

**Table 4.** Total energy per sector (million BOE) [31].

|  | Energy Type | Total |
| --- | --- | --- |
| 1 | Transportation | 388,417.946 |
| 2 | Industry | 317,568.463 |
| 3 | Household | 148,985.796 |
| 4 | Commercial | 43,484.632 |
| 5 | Other | 10,788.136 |
|  | Total | 909,244.973 |

Energy saving is currently being accomplished by accelerating the global energy transition, which is supported by a mutual agreement among all the International Energy Agency (IEA) members regarding energy efficiency. This acceleration can achieve the target of net-zero emissions in a global scope [61]. Indonesia has designed the implementation of energy management, especially in government regulations on energy conversion. Another effort made by the government is to expand the minimum performance standards (MEPS). Also, this regulation applies energy savings such as electric vehicles, induction cookers through the implementation of government programs, including diesel-to-gas generators, rooftop solar power plants, and electric motor conversion [62].

### 6.2.1. Transportation Sector

In 2021, the highest energy consumption rate was in the transportation sector (Figure 12), reaching around 388,417.946 million TOE. The percentage of fuel used in this sector is approximately 55.1%, while the remaining 44.23% and 0.07% are allocated to biogas, as well as gas and electricity, respectively. Of the total energy consumption, the use of gasoline RON 88-100 reached 54.32%, followed by gasoil, avtur and fuel oil at 3.29%, 3.08%, and 0.04%.

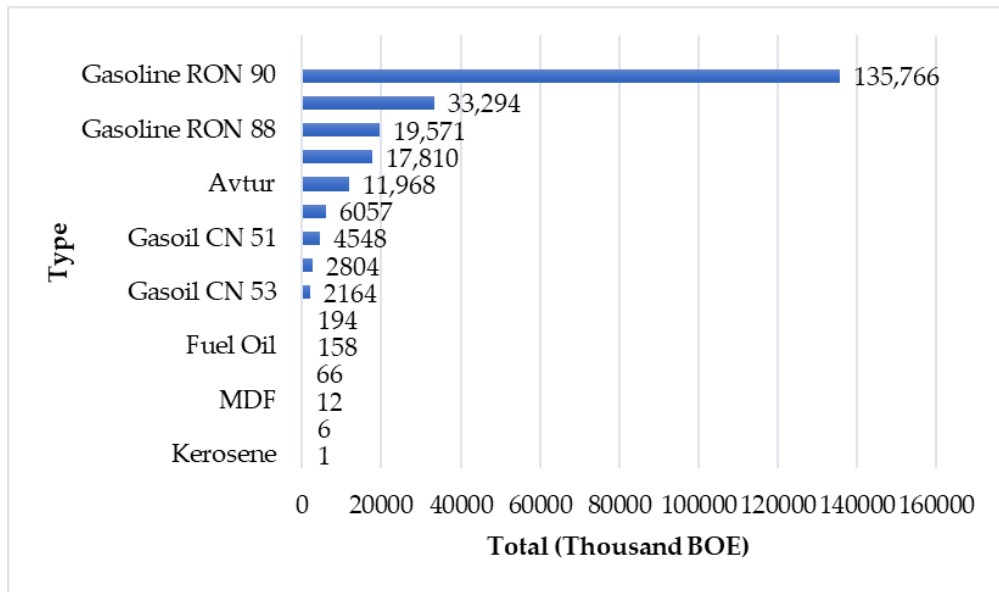

**Figure 12.** Energy consumption in the transportation sector [31].

To support a zero-emission scenario, at least 47% of a generation's shares need to come from renewable energy by 2030. In the next 10 years, solar PV capacity is anticipated to multiply a hundredfold to 108 GW. This is intended to usefully support increased electrification in the industrial and transportation sectors. The government has undertaken budget tagging to mark the utilization of public finance for climate change mitigation and adaptation, including energy and transportation activities. However, it had not been able to reduce emission and this in turn has an impact on budget allocations. In the past five years, the state budget has been used to fund the energy and transportation sectors, amounting to IDR 221.6 trillion (81.73%). The current budget allocations and expenditures still fall short of that needed to achieve the nationally determined contribution (NDC) targeted at IDR 318.18 trillion per year from 2020 to 2030 [63].

### 6.2.2. Industrial Sector

Energy consumption in the industrial sector without biomass was 317,568.463 million TOE, with the largest consumption of gas at 33.50%, followed by coal and electricity at 33.25% and 23.09%, respectively. However, in 2020, coal consumption decreased by 7.17%. Figure 13 is a detailed overview of energy use in the industrial sector.

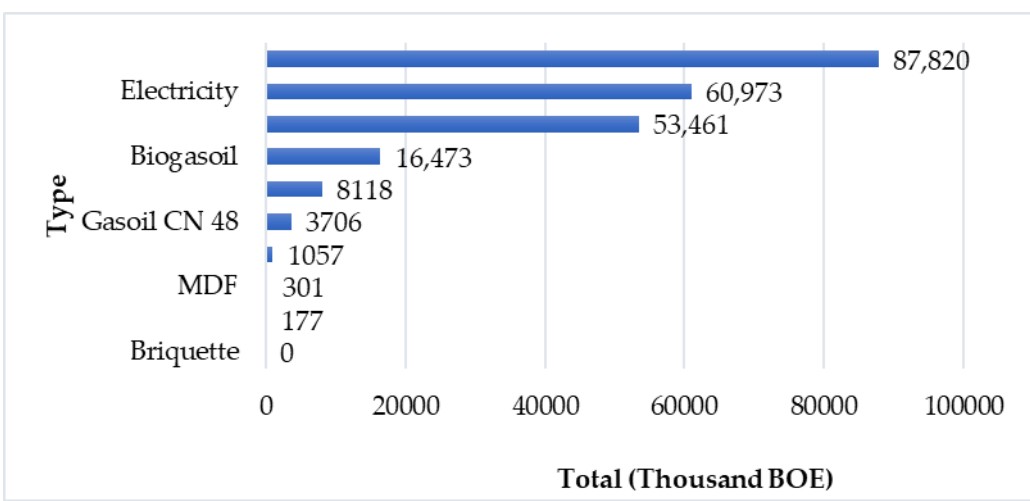

**Figure 13.** Energy consumption in the industrial sector [31].

Increased industrial and vehicular activities contributes significantly to the surge in energy demand in both sectors. In the industrial sector, the demand is projected to align with its growth as stipulated in "Indonesia Vision 2045". Meanwhile, in the transportation sector it is affected by the growing number of motor vehicles, the substitution program relating to the transformation from conventional to electric cars, the mandatory biodiesel and bioethanol initiatives as well as the shift from private to mass automobiles [64].

### 6.2.3. Household Sector

In 2021, energy consumption in households without biomass reached 148,986.796 million TOE consisting of electricity, LPG, kerosene, gas, and biogas at 49.03% and 48.78%, respectively, 1.85%, 0.21%, and 0.12% (Figure 14). The electricity is generally used for air conditioning (AC), washing machines, pumps, ironing, and lighting. Meanwhile, LPG is used for daily cooking, and kerosene is used in some remote areas for both cooking and lighting.

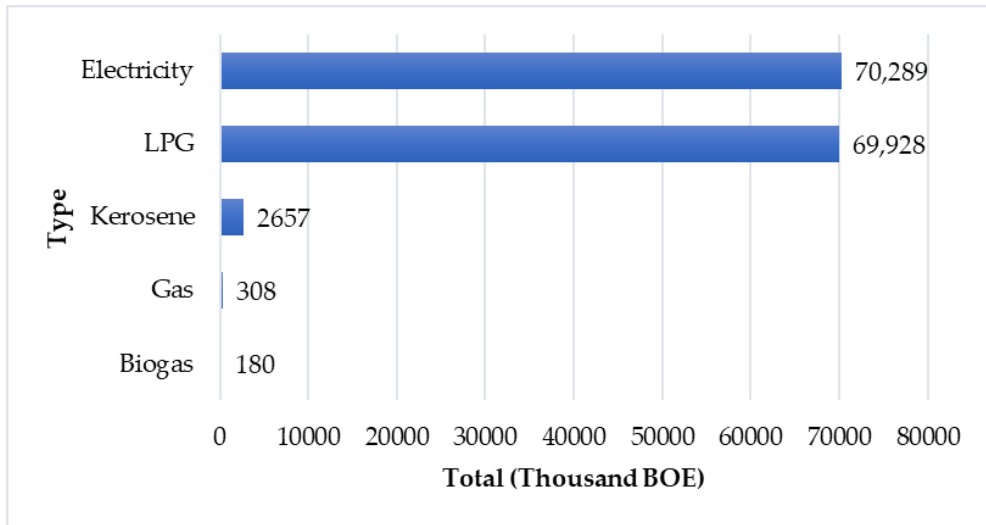

**Figure 14.** Energy consumption in households [31].

The program initiated by the government concerning the gas network construction for households was predicted to reach 4.7 million SR. Meanwhile, it is intended to be used as a reference in the projection of natural gas demand. In order to achieve the gas network development targeted for 2025, there is need to build approximately one million SR per year. In the BaU scenario, it is assumed to align with the country energy plan, while in the PB and RK, the growth is one million SR/year, and greater than one million/year, respectively. Based on the projection results, by 2050, natural gas demand in the BaU, PB, and RK scenarios is bound to reach 2.2 MTOE, 3.4 MTOE, and 4.5 MTOE, respectively [64].

### 6.2.4. Commercial Sector

Total energy consumption in the commercial sector without biomass, such as hotels, malls, hospitals and offices, is 43,485 million TOE with electricity, fuel, LPG, and gas values of 87.53%, 6.22%, 4.59%, and 1.66%. The electricity is used for cooling and lighting, while fuel is used for power plants. Meanwhile, gas and LPG are mostly used for cooking. The energy use in the commercial sector is shown in Figure 15.

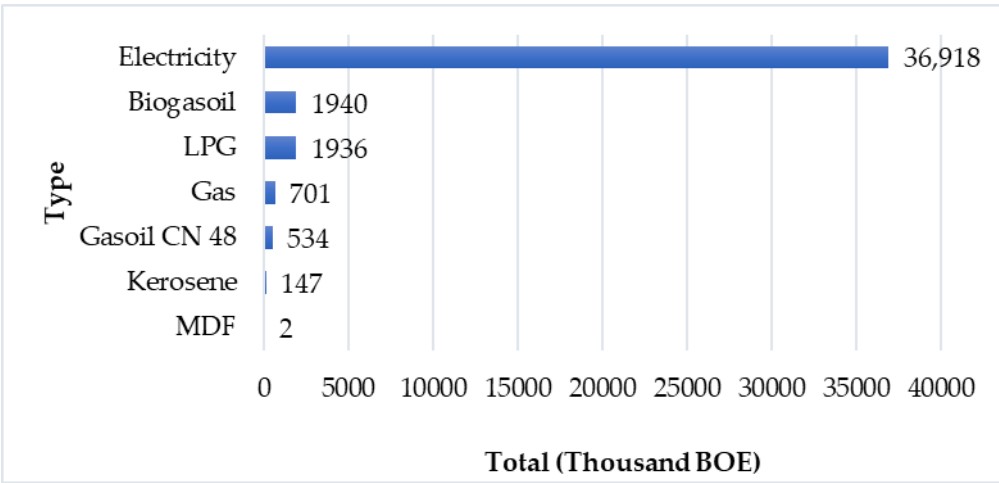

**Figure 15.** Energy consumption in commercial [31].

The large use of electricity is anticipated to affect the growth of electric vehicles by 2035, and this tends to follow a yearly trend. The government supports the application of induction cookers and the gas network development that encompasses the energy transition roadmap and neutral carbon by 2022. Furthermore, improving energy efficiency in commercial buildings should also consider its conservation from the design stage to building operations through the use of efficient equipment and systems [65].

### 6.2.5. Other Sectors

In 2021, other sectors consumed biodiesel, gasoline, fuel, diesel, and kerosene, reaching 10,788.136 million TOE (Figure 16). Generally, this energy is used for tractors, excavators, dump trucks, wheel loaders, belt conveyors, crushers, and generators to drive machinery in the construction sector.

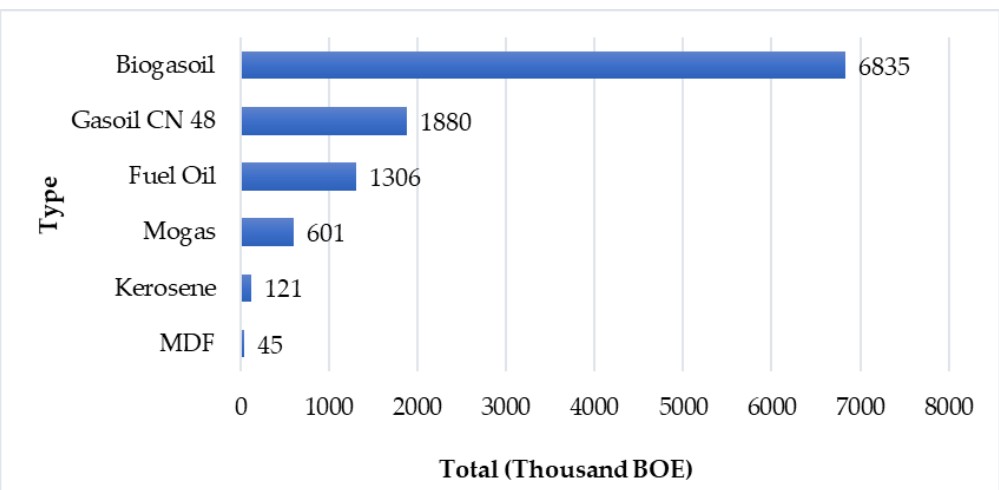

**Figure 16.** Energy consumption in other sectors [31].

The projected energy demand of other sectors, including the industrial field, is calculated based on the development of its GDP and their usage intensity. Data from the Ministry of Energy and Mineral Resources and Statistics Central Bureau are used to discern the intensity of other sectors. Meanwhile, the projected GDP growth of other sectors is assumed to be comparable to the industrial field [65].

## 7. Potential and Future Development

Indonesia is an archipelago with wealthy energy potentials spread across its 34 provinces. Therefore, a comprehensive map illustrating the technical potentials of renewable energy should be prepared. This also needs to support its transition towards the utilization of 100% renewable energy to achieve an emission-free Indonesia by 2050 [66]. The development of renewable energy is possible for the current geographical conditions; some power plants have been developed by the government and installed in several places. The population and economy have also grown exponentially, and presently this has an impact on climate, ecosystem processes, and biodiversity. The indicators of socioeconomic status and other ecological impacts tend to correlate with energy usage. Energy is the fuel used for global economic activities, such as population expansion, improved quality of life and growth in consumption. In Indonesia, fossil fuel in the form of oil, gas, and coal is still being used. Its massive use drives economic growth, although it is often accompanied by ecological damage that brings about potential natural or man-made disasters.

In respect to the Nationally Determined Contribution (NDC), all the countries worldwide, including Indonesia, were committed to maintaining a global temperature rise of 1.5 °C to 2 °C, in the first period. This is aimed to reduce emissions by 29% based on personal efforts and 41% assuming there is international cooperation. The no-action initiative, anticipated to be implemented in 2030, will be realized through the forestry sector, energy including transportation, waste, industrial processes, and product use, as well as agriculture. This commitment is strengthened by the government law on the ratification of the Paris Agreement on the United Nations Framework Convention on Climate Change (UNFCC). To achieve this, Indonesia set a renewable energy target in the national energy mix of at least 23% and 31% by 2025 and 2050, respectively. In addition, it has all the potentials for renewable energy, such as solar, hydro, wind, geothermal, and bioenergy or biomass [67].

Although Indonesia has a high potential for renewable energy at 419 GW, its utilization is still minimal, as shown in Table 5. Therefore, optimizing the use of renewable energy for power generation is part of the strategic plan for the development of NZE [68,69].

**Table 5.** Energy potential in Indonesia [18].

| Energy Type | Potency (GW) |
| --- | --- |
| Hydro energy | 75 |
| Geothermal | 23.7 |
| Bioenergy | 32.6 |
| Solar | 207.8 |
| Wind | 60.6 |
| Micro-hydro | 19.3 |
| Total | 419 |

The lack of renewable energy utilization for electricity is due to the relatively high production price of the plants. This makes it difficult to compete with fossil plants, especially coal. Moreover, lack of domestic industrial support and difficulty in obtaining low-interest funding are also some causes that tend to obstruct renewable energy development.

### 7.1. Hydro Energy

Hydro is a type of reliable power plant first built in 1938 during the Dutch era and known as the Jelok Hydro Power Plant. Figure 17 shows that the micro-hydro power plant has a potential of 95 GWh per year [70], consisting of hydro of 75,000 MW and a micro-hydro of 19,370 MW [18]. However, according to the updated data report in the fourth quarter of 2021, the potential for newly utilized hydro was 6601.80 MW, which included a hydro power plant of 5638.7 MW and micro-hydro of 126.4 MW and mini-hydro of 375.5 MW or approximately 6.5% of the potential of the hydro power plant [50].

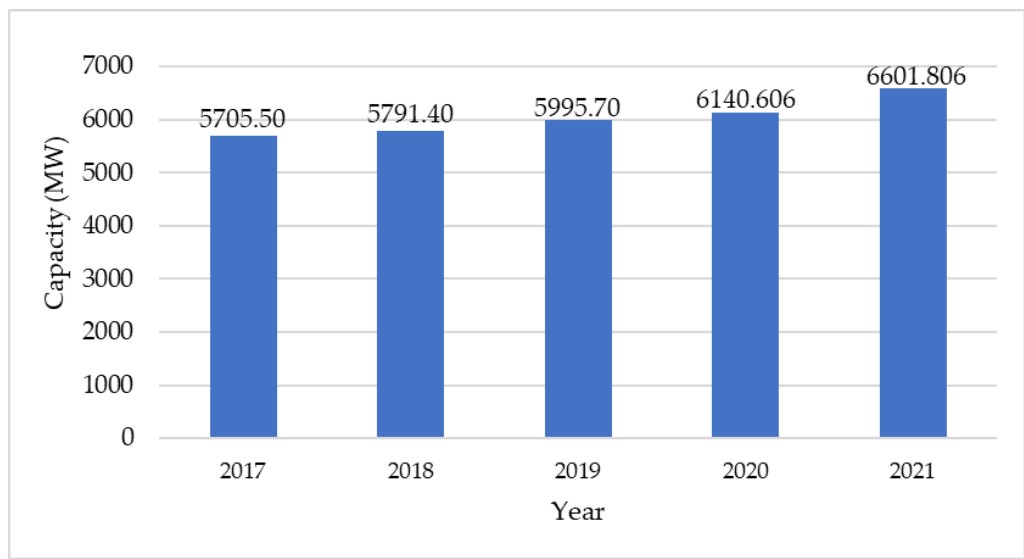

**Figure 17.** Installed capacity of hydro power plant up to the fourth quarter 2021 [50].

The target for constructing a hydro power plant will be 21.9 GW in 2030 [66]. Meanwhile, the national energy strategy contains a roadmap for renewable energy plants, which targets at building a 7.7 GW by 2030. In Indonesia, almost all provinces have hydro energy potential, with the distribution shown in Table 6.

**Table 6.** Potential of hydro energy in every province [18].

| No. | Island | Hydro (GW) | Micro-Hydro (GW) |
|---|---|---|---|
| 1. | Sumatra | 15.6 | 5.73 |
| 2. | Java | 4.2 | 2.91 |
| 3. | Kalimantan | 21.6 | 8.1 |
| 4. | Sulawesi | 10.2 | 1.67 |
| 5. | Bali and Nusa Tenggara | 0.62 | 0.14 |
| 6. | Maluku | 0.43 | 0.21 |
| 7. | Papua | 22.35 | 0.62 |
| | Total | 75.00 | 19.37 |

The utilization of hydro potential into electricity based on its scale is grouped in three, namely hydro, micro-hydro and mini-hydro power plants. The target for an additional installed capacity of micro-hydro in 2021 was 557.9 MW, and the realization of installed capacity by the fourth quarter was 461.19 MW with details of 350 MW hydropower and 111.19 MW mini-hydro.

In 2022, a hydro power plant with a capacity of 515 MW in Poso, Central Sulawesi, was started. In the same year, the Jatigede Plant, which was supposed to operate in 2021, but due to geological constraints, started operation. The plant of Peusangan 1 is targeted to operate in July 2023, and then in 2024, the Asahan 3 will also be able to operate.

Hydro Dam Potential

Indonesia has a large hydro dam potential with a total of 95,003 MW. This can support the development of the renewable energy-based industry program for hydro power plants [71]. The Cipanas dam, located in Cibuluh, Sumedang, with a size of 9243 ha and a height of 65 m, is commonly used for irrigation. Based on the calculations, this power plant has a 2 × 1.8 MW or 3.6 MW capacity and production energy of 28,309.7 MWh per year [72]. Meanwhile, Bagong dam in Sumurup and Sengon, Trenggalek, with a height of

82 m, the capacity of 15.5 million m$^3$ and a water supply of 0.3 m$^3$ has the potential power of 0.52 MW.

The Tugu dam, located in Nglinggis Village, Tugu sub-district, Trenggalek, with an area of 104 ha, depth of 27.85 m, and height of 81 m, has a power of 0.4 MW. Hence, on average, the two dams can produce electrical power of 0.92 MW, which is 22.08 MWh per day [73]. Furthermore, the Merangin dam has been through a simulation of the operation pattern of the hydro power plant for 19 years. Hydrological, HEC-HMS model generation and technical data on dam planning had average power gain, Pb, and total energy of 103.8 MW, 98.53 MW, and 636.66 GWh/year [74].

### 7.2. Geothermal

Geothermal energy started developing 100 years ago with the first geothermal well drilled in Kamojang by the Dutch in 1926 and has been operational since 1983 [75,76]. It has a geothermal potential of 23,766 MW, with the distribution as shown in Table 7.

**Table 7.** Distribution of geothermal potential in 2021 [50].

| No. | Location | Potency (MW) |
|---|---|---|
| 1. | Sumatra | 9517 |
| 2. | Java | 8050 |
| 3. | Bali | 335 |
| 4. | Nusa Tenggara | 1399 |
| 5. | Kalimantan | 175 |
| 6. | Sulawesi | 3071 |
| 7. | Maluku | 1144 |
| 8. | Papua | 75 |
| Total | | 23,766 |

As shown in Figure 18, the installed capacity based on updated data in the fourth quarter of 2021 was 2185.7 MW, which means a lot of geothermal energy has not been utilized. Therefore, the government targets an increase in geothermal utilization to 7241.5 MW or 16.8% in 2025 [66]. The installed capacity of geothermal power plant was 2185.7 MW until the fourth quarter of 2021, when an additional 155.35 MW, comprising 56.95 MW and 98.4 MW in Sorik Merapi Unit 2 and Rantau Dedap Unit 1, were added.

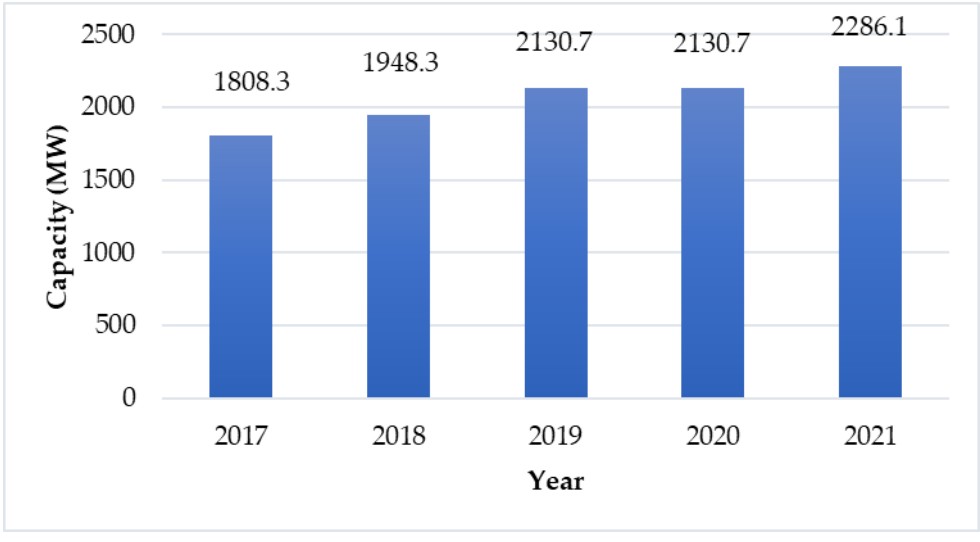

**Figure 18.** Installed capacity of geothermal power plant up to the fourth quarter of 2021 [50].

### 7.3. Bioenergy

In addition to the electricity sector, bioenergy as a renewable energy source can also be used to meet energy needs in the transportation, industry, and household sectors [77]. The diversity of raw materials such as livestock manure, agricultural, plantation, and urban waste is easy to discover. This energy can be used as a power plant for biomass, biogas, municipal waste, household biogas, bioenergy furnaces, etc.

Table 8 shows the distribution of bioenergy potentials in 34 provinces, and in addition to its amount is also affected by the diverse regions.

**Table 8.** Bioenergy potential in Indonesia in 2021 [50].

| No. | Province | Area (km²) | Technical Potential (MW) | No. | Province | Area (km²) | Technical Potential (MW) |
|---|---|---|---|---|---|---|---|
| 1. | Riau | 87,023.66 | 4195.1 | 18. | West Nusa Tenggara | 18,572.32 | 394.1 |
| 2. | East Java | 47,803.49 | 3420.9 | 19. | Lampung | 34,623.80 | 7763 |
| 3. | North Sumatra | 72,981.23 | 2911.6 | 20. | Central Sulawesi | 61,841.29 | 326.9 |
| 4. | West Java | 35,377.76 | 2554,1 | 21. | East Nusa Tenggara | 48,718.10 | 240.5 |
| 5. | Central Java | 32,800.76 | 2232.5 | 22. | D.I. Yogyakarta | 3133.15 | 224.2 |
| 6. | South Sumatra | 91,592.43 | 2132.6 | 23. | Bangka Belitung | 16,424.06 | 223.1 |
| 7. | Jambi | 50,058.16 | 1839.9 | 24. | West Sulawesi | 16,787.18 | 205.9 |
| 8. | Central Kalimantan | 153,564.50 | 1498.9 | 25. | Bali | 5780.06 | 191.6 |
| 9. | Lampung | 34,623.80 | 1492.1 | 26. | North Sulawesi | 13,892.47 | 164.0 |
| 10. | West Kalimantan | 147,307.00 | 1308.2 | 27. | Southeast Sulawesi | 38,067.70 | 1677 |
| 11. | South Kalimantan | 38,744.23 | 1289.9 | 28. | Gorontalo | 11,257.07 | 130.6 |
| 12. | Aceh | 57,956.00 | 1174.3 | 29. | Jakarta | 664.01 | 126.6 |
| 13. | East/North Kalimantan | 129,066.64 | 964.3 | 30. | Papua | 319,036.05 | 96.5 |
| 14. | South Sulawesi | 46,717.48 | 959.4 | 31. | West Papua | 102,955.15 | 54.9 |
| 15. | West Sumatra | 42,012.89 | 957.8 | 32. | North Maluku | 31,982.50 | 34.5 |
| 16. | Bengkulu | 19,919.33 | 644.8 | 33. | Maluku | 46,914.03 | 32.6 |
| 17. | Banten | 9662.92 | 465.1 | 34. | Riau Islands | 8201.72 | 15.9 |
| | Total | | | | | | 32,653.8 MW |

Until the fourth quarter of 2021 as shown in Figure 19, bioenergy power plants installed capacity was 2284 MW [50]. This was in addition to 19.5 MW consisting of 8.5 MW, 2 MW and 9 MW biomass, biogas, and waste power plants. The national energy plan is targeted to reach 9.6 GW from bioenergy power plants and 1.09 GW from biomass, biogas, and waste bioenergy.

The abundant potential of bioenergy opens wider opportunities for the younger generation to contribute directly to efforts for the development of bioenergy and clean energy through various lines [78]. In the academic sector, research and innovation development opportunities must be explored more deeply to maximize the domestic bioenergy potential. Community service programs organized by universities can also contribute to innovations in the daily use of bioenergy at the community level.

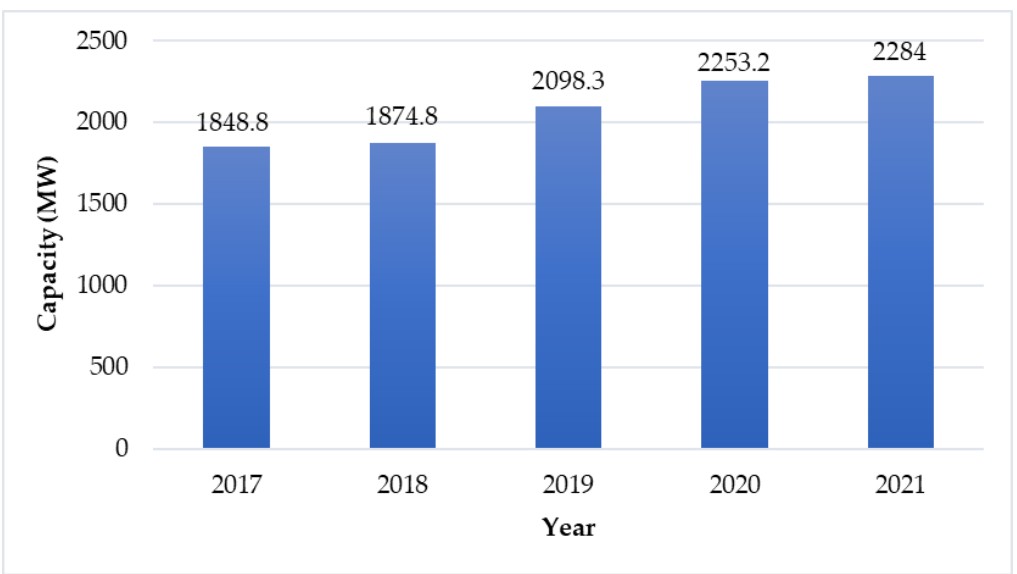

**Figure 19.** Installed bioenergy power plant until the fourth quarter of 2021 [50].

### 7.3.1. Biomass

Biomass is material from living organisms, including plants, animals, and their by-products [30]. In Indonesia, it can be extracted from industrial waste such as palm oil, tapioca, pulp and paper, sugar cane, rice, and wood. The biomass potential for electricity is 32,654 MW with an installed capacity of 151.52 MW on-grid and 1969.64 MW off-grid in 2021 [79]. The prospect of biomass energy in the future is that it can be utilized by the co-firing method by mixing with coal in a steam power plant. In this co-firing, the raw materials are waste and wood.

### 7.3.2. Biogas

The potential for biogas in the country is 2602.6 MW. The capacity of on-grid and off-grid biogas power plants in 2021 was 22.10 MW and 112.69 MW [79]. In addition to being able to be used as a power plant, biogas can also be used for households by utilizing cow dung and household waste, known as communal biogas development [80]. Based on data on the implementation of biogas development, the realization of biogas performance achievements until the fourth quarter of 2021 was 28,392 thousand $m^3$, which is 101.90% of the target.

### 7.3.3. Biofuel

Generally, biofuels contain energy and components obtained from plants and biomass. Research on biofuels produced from biomass resources by environmentally friendly methods keeps increasing [81]. Various liquid and gaseous biofuels can be produced from biomass, such as ethanol, biodiesel, methane, methanol, and bio-oil [82]. To increase the use of renewable energy, efforts have been made to mix palm oil with diesel oil to produce biodiesel. Furthermore, efforts have also been made to mix ethanol from sugarcane processing with gasoline to produce bioethanol.

The realization of biofuel utilization until the fourth quarter of 2021 has reached 6.66 million kL of the initial target of 10.2 million kL for domestic usage. However, there is an adjustment to the 2021 target to 9.2 million kL. The prospect of biofuel in the future is that it can be a substitute for petroleum; hence, those derived from land or marine plants, such as microalgae, have started to be pursued as a source of alternative energy.

### 7.3.4. Waste

The waste produced by the community can be one of the energy sources that can produce approximately 2000 MW [83]. Currently, the operational waste power plant located

in Benowo, Surabaya has a capacity of 12 MW, which was started on 6 May 2021. The prospect is quite good, with several locations including Surabaya, Jakarta, Tangerang, Bandung, Semarang, Surakarta, Makassar, Denpasar, Manado, Palembang, Bekasi, and South Tangerang City. The development plan is 9 MW, 10 MW, 20 MW, 20 MW, 38 MW, and 29 MW in Bekasi City, Surakarta, Palembang, Denpasar, Jakarta, and Bandung. The three remaining cities, including Makassar, Manado, and South Tangerang have the same capacity of 20 MW each [18].

### 7.4. Solar Energy

One source of renewable energy that is developing quite rapidly all over the world is solar energy. As a tropical country that obtains sunlight throughout the year, energy needs to be optimized [84]. The potential for developing solar energy (Table 9) is very large with 207,898 MW [85–88] and an average solar light intensity of 4.80 kWh/m$^2$/day [85,89]. The availability of solar potential is a necessary first step in the utilization of solar energy in Indonesia.

**Table 9.** Potential of solar energy in 34 provinces of Indonesia [83].

| No. | Province | Area (km$^2$) | Technical Potential (MW) | No. | Province | Area (km$^2$) | Technical Potential (MW) |
|---|---|---|---|---|---|---|---|
| 1. | Aceh | 57,956.00 | 7881 | 18. | Riau Islands | 8201.72 | 753 |
| 2. | Bali | 5780.06 | 1254 | 19. | Lampung | 34,623.80 | 7763 |
| 3. | Bangka-Belitung | 16,424.06 | 2810 | 20. | Maluku | 46,914.03 | 2238 |
| 4. | Banten | 9662.92 | 2461 | 21. | North Maluku | 31,982.50 | 2020 |
| 5. | Bengkulu | 19,919.33 | 3475 | 22. | West Nusa Tenggara | 18,572.32 | 3036 |
| 6. | D.I. Yogyakarta | 3133.15 | 996 | 23. | East Nusa Tenggara | 48,718.10 | 9931 |
| 7. | DKI Jakarta | 664.01 | 225 | 24. | Papua | 319,036.05 | 7272 |
| 8. | Gorontalo | 11,257.07 | 1218 | 25. | West Papua | 102,955.15 | 2035 |
| 9. | Jambi | 50,058.16 | 8847 | 26. | Riau | 87,023.66 | 6307 |
| 10. | West Java | 35,377.76 | 9099 | 27. | West Sulawesi | 16,787.18 | 1677 |
| 11. | Central Java | 32,800.76 | 8753 | 28. | South Sulawesi | 46,717.48 | 7588 |
| 12. | East Java | 47,803.49 | 10,335 | 29. | Central Sulawesi | 61,841.29 | 6186 |
| 13. | West Kalimantan | 147,307.00 | 20,113 | 30. | Southeast Sulawesi | 38,067.70 | 3917 |
| 14. | South Kalimantan | 38,744.23 | 6031 | 31. | North Sulawesi | 13,892.47 | 2113 |
| 15. | Central Kalimantan | 153,564.50 | 8459 | 32. | West Sumatera | 42,012.89 | 5898 |
| 16. | East Kalimantan | 129,066.64 | 13,479 | 33. | South Sumatera | 91,592.43 | 17,233 |
| 17. | North Kalimantan | 75,467.70 | 4643 | 34. | North Sumatera | 72,981.23 | 11,851 |
| | | | Total | | | | 207,898 MW |

The utilization of solar energy is accomplished by the production of on-grid and off-grid energy with a total installed capacity of 190.15 MW in 2021 (Figure 20). The data was updated in the fourth quarter of 2021 with the addition of 31.45 MWp capacity, consisting of solar power for customers' roofs of 27.4 MWp and ground mounted solar power plants of 4.05 MWp [50]. Until the fourth quarter of 2021, solar ground mounted with commercial operation date (COD) was on off-grid Papagarang and Sei Make.

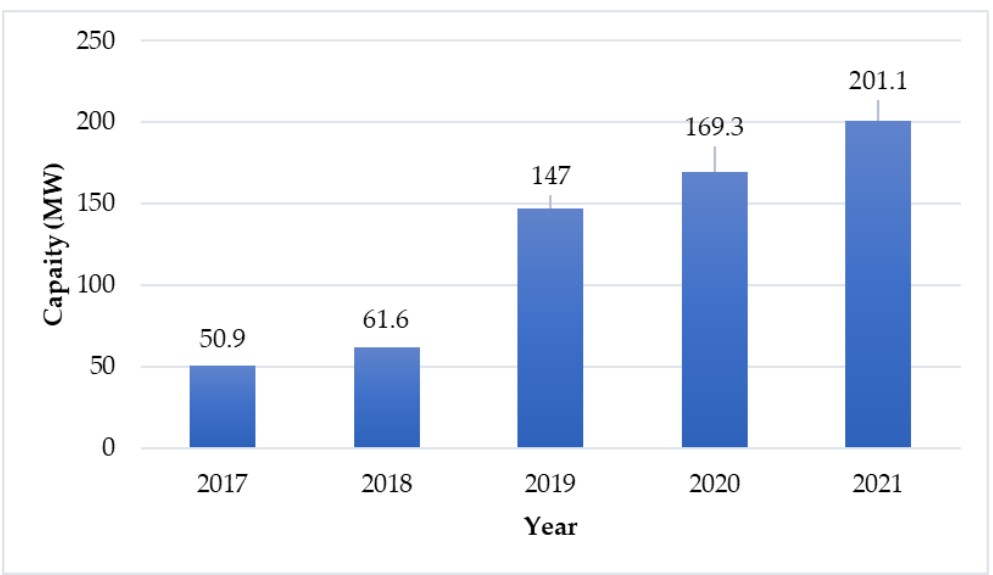

**Figure 20.** Solar power plant installed capacity until the fourth quarter of 2021 [50].

The solar power plant is targeted to reach 6.5 GW in 2025 and 14.2 GW by 2030. Meanwhile, the installed capacity is expected to reach 11.2 GW in 2030. The solar power plant is intermittent, meaning the energy is unstable, with the output dependent on seasonal conditions, humidity, temperature, cloud movement, and other weather conditions. This makes the generator to be unable to operate continuously at its installed capacity. Furthermore, the construction is also related to the high investment costs, hence the selling price of electricity is not economical.

In 2019, the government issued the regulation concerning the use of rooftop solar plant by consumers. This regulation was intended to open up opportunities for all consumers from the household sector, business, social government, and industry to participate in the utilization and management of renewable energy to achieve energy security and independence.

In addition to the rooftop solar power plant, solar energy in the future can be used as energy saving solar lamps. Furthermore, the government has accelerated the implementation of providing energy-efficient solar lights for people who do not have access to electricity. This policy is related to the distribution of energy saving solar lights in the border, underdeveloped, and isolated areas far from the electricity network.

The next prospect is the construction of a solar streetlight, a light that uses sunlight as a source of electrical energy. From 2016 to 2020, 65,501 solar street light units were built, with 18,888 units installed [18]. Meanwhile, in 2021, 4829 units of quarter four were installed with a focus on road locations without access to the electricity network. In 2020, the solar plant installation work was divided into two, namely the installation on rooftops and in cold storage facilities. It can also be applied in buildings, both as the main source and as a backup from existing power sources. One of the uses of electricity from solar plant is cold storage, and in 2021, 100 units comprising 88 units of rooftop and 12 units were found in cold storage public facilities.

Potential Map of Floating Solar Power Plant

Apart from being able to be applied on land, a solar plant can also be applied in water in accordance with the condition of Indonesia as an archipelagic country. This is also a large potential for solar energy in a tropical country. Subsequently, a floating solar power plant was made and placed on water bodies such as lakes, reservoirs, etc. The components include solar modules, platforms, pontoons, mooring systems, inverters, power conditions stations, cabling, network interconnection infrastructure, supporting facilities, meteorological centers, remote monitoring, and data collection systems (Figure 21).

It has more challenges than a common solar power plant on land due to the lack of track record, the uncertainty of costs, and its impact on the environment. This model is also relatively complicated in designing, building, and operating because it is related to electrical, anchoring, and mooring systems [90].

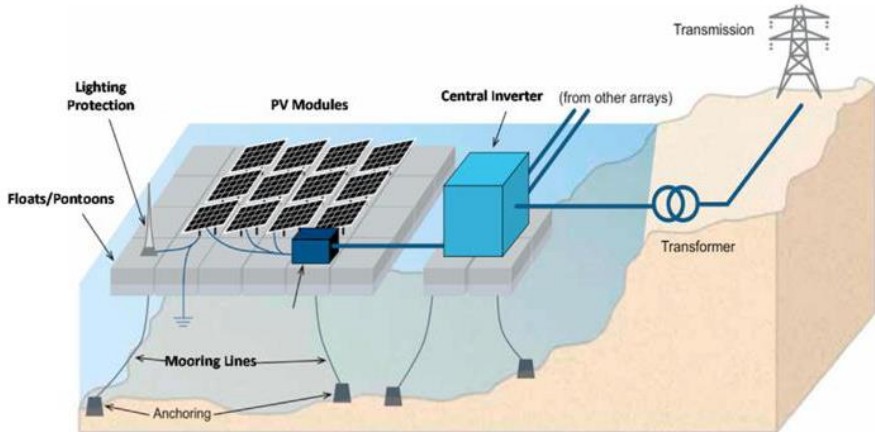

**Figure 21.** Example of a floating solar plant layout and its constituent components [90].

A floating solar plant has many advantages, such as not requiring land, which is generally valuable, reducing the occurrence of water evaporation, and inhibiting the growth of other weeds such as water hyacinth and evaporation. Furthermore, it is held back by the PV module creating a cooling system, which increases the efficiency of electricity generated. Information on the location, potential lakes, and areas (ha) spread across Indonesia is presented in the Ministry of Energy and Mineral Resources One Map [71]. Currently, the Citara floating plant is estimated to be completed in 2022, with another built in the Sutami Reservoir, Malang, in 2023.

In Central Java province, the increase in renewable energy mix was predicted to reach 21.82% by 2025, especially in solar and geothermal potentials. This province has an average solar potential of 4.05 kWh/kWp per day (greater than the national average of 3.75 kWh/kWp per day). Based on the national data sources obtained (Pusdataru Central Java and WRDC), there are 43 reservoirs located in Central Java except for the Kedunguling, which was excluded because it was experiencing drought. Incidentally, of the 42 artificial reservoirs, the potential for floating solar power plants is 727.25 MWp. A total of 92.3% (or 671.85 MWp) was contributed by the technicality of large reservoirs (11 reservoirs), 7.36% (53.25 MWp) from medium types (24 reservoirs), and the rest (2.14 MWp) was realized by the small ones (7 reservoirs) [91]. The potentials of the floating solar power plant in Central Java are shown in Tables 10–12.

**Table 10.** Technical potential of floating solar power plant in large reservoirs (>100 ha) [91].

| No. | Reservoir | Year Built | Surface Area Reservoir | | Potential Floating Solar Power Plant (MWp) | Generation Potential (GWh/Years) |
|---|---|---|---|---|---|---|
| | | | Total (ha) | 5% Area (ha) | | |
| 1 | Kedung Ombo | 1985–1989 | 4600 | 230.00 | 267.95 | 367.39 |
| 2 | Gajahmungkur | 1976–1982 | 2539 | 126.90 | 147.88 | 201.67 |
| 3 | Wadaslintang | 1983–1987 | 1320 | 66.00 | 76.89 | 90.26 |
| 4 | Mrica | 1984–1989 | 1250 | 62.50 | 72.80 | 87.36 |
| 5 | Cacaban | 1951–1959 | 790 | 40.00 | 46.02 | 57.48 |
| 6 | Sempor | 1974–1978 | 275 | 13.75 | 16.10 | 19.38 |
| 7 | Cengklik | 1923–1928 | 253 | 12.65 | 14.80 | 23.94 |
| 8 | Jombor | 1920 | 164 | 8.21 | 9.61 | 14.61 |
| 9 | Penjalin | 1930–1934 | 120 | 6.00 | 6.98 | 9.92 |
| 10 | Jatibarang | 2009–2014 | 111 | 5.55 | 6.40 | 10.00 |
| 11 | Gembong | 1930–1933 | 110 | 5.50 | 6.41 | 9.48 |
| | Total | | | | 671.85 MWp | 891.5 GWh/years |

**Table 11.** Technical potential of floating solar power plant in medium reservoirs (10–100 ha) [91].

| No. | Reservoir | Year Built | Surface Area Reservoir | | Potential Floating Solar Power Plant (MWp) | Generation Potential (GWh/Years) |
|---|---|---|---|---|---|---|
| | | | Total (ha) | 5% Area (ha) | | |
| 1 | Logung | 2014–2018 | 88.50 | 4.43 | 5.15 | 7.54 |
| 2 | Malahayu | 1935–1940 | 70.00 | 3.50 | 4.08 | 6.13 |
| 3 | Klego | 1943 | 68.60 | 3.43 | 3.97 | 6.49 |
| 4 | Garung | 1978–1983 | 67.00 | 3.35 | 3.90 | 5.14 |
| 5 | Lalung | 1940 | 63.96 | 3.20 | 3.71 | 5.76 |
| 6 | Lodan Wetan | 1994 | 60.54 | 3.03 | 3.40 | 5.20 |
| 7 | Mulur | 1918–1926 | 59.40 | 2.97 | 3.40 | 5.23 |
| 8 | Greneng | 1919 | 51.00 | 2.55 | 2.91 | 4.32 |
| 9 | Delingan | 1920–1923 | 47.00 | 2.35 | 2.73 | 4.14 |
| 10 | Ketro | 1975–1984 | 47.00 | 2.35 | 2.73 | 4.35 |
| 11 | Gunung Rowo | 1918–1925 | 44.28 | 2.21 | 2.56 | 3.77 |
| 12 | Krisak | 1942–1943 | 44.00 | 2.20 | 2.52 | 3.74 |
| 13 | Banyukuwung | 1996 | 34.94 | 1.75 | 2.08 | 3.07 |
| 14 | Ngancar | 1944–1946 | 34.00 | 1.70 | 1.90 | 2.71 |
| 15 | Grawan | 2004 | 17.92 | 0.90 | 1.05 | 1.54 |
| 16 | Nglangon | 1911–1914 | 17.00 | 0.85 | 0.97 | 1.50 |
| 17 | Parangjoho | 1973–1980 | 16.72 | 0.84 | 0.96 | 1.41 |
| 18 | Panohan | 2005–2009 | 16.20 | 0.81 | 0.95 | 1.42 |
| 19 | Plumbon | 1918–1928 | 13.75 | 0.69 | 0.80 | 1.17 |
| 20 | Botok | 1942 | 13.44 | 0.67 | 0.78 | 1.20 |
| 21 | Kembangan | 1939–1940 | 13.40 | 0.67 | 0.77 | 1.18 |
| 22 | Sanggeh | 1909–1911 | 13.00 | 0.65 | 0.75 | 1.16 |
| 23 | Nawangan | 1974–1976 | 10.40 | 0.52 | 0.60 | 0.87 |
| 24 | Gebyar | 1944–1945 | 10.00 | 0.50 | 0.58 | 0.89 |
| | Total | | | | 53.25 MWp | 79.93 GWh/years |

**Table 12.** Technical potential of floating solar power plant in small reservoirs (<10 ha) [91].

| No. | Reservoir | Year Built | Surface Area Reservoir | | Potential Floating Solar Power Plant (MWp) | Generation Potential (GWh/years) |
|---|---|---|---|---|---|---|
| | | | Total (ha) | 5% Area (ha) | | |
| 1 | Butak | 1901–1902 | 8.39 | 0.42 | 0.47 | 0.73 |
| 2 | Songputri | 1977–1984 | 8.20 | 0.41 | 0.47 | 0.68 |
| 3 | Simo | 1904–1907 | 6.67 | 0.33 | 0.37 | 0.58 |
| 4 | Pejengkolan | 1984–1986 | 4.85 | 0.24 | 0.28 | 0.42 |
| 5 | Tempuran | 1914–1916 | 4.72 | 0.24 | 0.27 | 0.39 |
| 6 | Brambang | 1939–1940 | 4.05 | 0.20 | 0.23 | 0.36 |
| 7 | Blimbing | 1922–1922 | 1.07 | 0.05 | 0.06 | 0.09 |
| | Total | | | | 2.14 MWp | 3.24 GWh/years |

In the large reservoir category, Kedung Ombo, Gajah Mungkur, Wadaslintang, and Mrica have the greatest potentials, of 267.95, 147.88, 76.89, and 72.80 MWp, respectively. Meanwhile, in the medium reservoir category, Jatibarang, Logung, and Malahayu have the third largest potentials, with a capacity of approximately 5 MWp each. For the small reservoir category, the seven of them have a capacity of less than 0.5 MWp. Supposing all these reservoirs are fitted with floating solar power plants with these potential capacities, the resulting electricity generation was predicted to reach 974.66 GWh per year or relatively 3.47% of the net electricity production in Central Java and Yogyakarta by 2018 [91].

### 7.5. Wind Power Plant

Wind power plant is targeted to reach 7 GW in 2030 with an installed capacity of 2.2 GW. Several areas in Indonesia have wind potential with a speed of 4 m/s–6 m/s. Table 13 shows wind potential per province.

**Table 13.** Wind power plant potential in 34 provinces of Indonesia [66].

| No. | Province | Area (km²) | Technical Potential (MW) | No. | Province | Area (km²) | Technical Potential (MW) |
|---|---|---|---|---|---|---|---|
| 1. | East Nusa Tenggara | 48,718.10 | 10,188 | 18. | Riau Islands | 8201.72 | 922 |
| 2. | East Java | 47,803.49 | 7907 | 19. | Central Sulawesi | 61,841.29 | 908 |
| 3. | West Java | 35,377.76 | 7036 | 20. | Aceh | 57,956.00 | 894 |
| 4. | Central Java | 32,800.76 | 5213 | 21. | Central Kalimantan | 153,564.50 | 681 |
| 5. | South Sulawesi | 46,717.48 | 4193 | 22. | West Kalimantan | 147,307.00 | 554 |
| 6. | Maluku | 46,914.03 | 3188 | 23. | West Sulawesi | 16,787.18 | 514 |
| 7. | West Nusa Tenggara | 18,572.32 | 2605 | 24. | North Maluku | 31,982.50 | 504 |
| 8. | Bangka Belitung | 16,424.06 | 1787 | 25. | West Papua | 102,955.15 | 437 |
| 9. | Banten | 9662.92 | 1753 | 26. | West Sumatra | 42,012.89 | 428 |
| 10. | Bengkulu | 19,919.33 | 1513 | 27. | North Sumatra | 72,981.23 | 356 |
| 11. | Southeast Sulawesi | 38,067.70 | 1414 | 28. | South Sumatra | 91,592.43 | 301 |
| 12. | Papua | 319,036.05 | 1411 | 29. | East Kalimantan | 129,066.64 | 212 |
| 13. | North Sulawesi | 13,892.47 | 1214 | 30. | Gorontalo | 11,257.07 | 137 |
| 14. | Lampung | 34,623.80 | 1137 | 31. | North Kalimantan | 75,467.70 | 73 |
| 15. | D.I. Yogyakarta | 3133.15 | 1079 | 32. | Jambi | 50,058.16 | 37 |
| 16. | Bali | 5780.06 | 1019 | 33. | Riau | 87,023.66 | 22 |
| 17. | South Kalimantan | 38,744.23 | 1006 | 34. | DKI Jakarta | 664.01 | 4 |
| | Total | | | | | | 60,647 MW |

According to updated data in the fourth quarter of 2021, Indonesia has an installed wind plant of 154.3 MW, while the target capacity in 2025 is 255 MW. Therefore, the country contains two large plants, namely Sidrap and Tolo. Sidrap is located in Sidenreng Rappang with 30 windmills at a capacity of 75 MW. Tolo is located in Turatea, in South Sulawesi, and has a capacity of 72 MW with 20 wind turbines comprising 3.6 MW each.

The prospect of wind energy is also fairly good, with the future possibility of building plant in South Garut in three sub-districts, namely Pameungpeuk, Cibolang, and Cisompet. Several plants will also be built in 2023, such as the Sukabumi Project and the Tolo II in Jeneponto.

### 7.6. Nuclear Power

Two of the basic raw materials in nuclear manufacture are uranium and thorium, which are radioactive elements. Indonesia has a total of 81,091 tons of uranium resources and thorium deposits of 140,411 tons. Table 14 shows the potential for uranium and thorium in Indonesia [18].

**Table 14.** The potential for uranium and thorium in Indonesia [18].

| Region | Uranium (Ton) | Thorium (Ton) |
|---|---|---|
| Sumatra | 31,567 | 126,821 |
| Kalimantan | 45,731 | 7028 |
| Sulawesi | 3793 | 6562 |
| Total | 81,091 | 140,411 |

According to 2015 data, the country's thorium and uranium reserves are 130,974 tons and 74,397 tons. In addition to the Babylon Islands, thorium potential is also found on Singkep Island, West Kalimantan, and Mamuju.

Indonesia has the potential to build nuclear power plants to fulfil domestic needs with the help of the economic and industrial sectors. It can be the first country in Southeast Asia to have a nuclear power plant due to the availability of uranium, which has the potential to become a major export source. In terms of impact, this energy can overcome the waste produced without affecting electricity costs. The outcome, which cannot pollute the environment, is deposited into the ground because it will not negatively affect the surrounding community [92].

## 8. Conclusions

In conclusion, renewable energy is sustainable, affordable, reliable, and a safer means of supplying electricity for social and economic infrastructure development. The Indonesian government has developed project on the use of renewable and sustainable energy. Therefore, this paper recommends the extraction of large-scale energy by properly utilizing hydro, geothermal, solar, bioenergy, and wind energy sources. The project is supported through reforestation, prevention of deforestation, increasing renewable energy capacity, and energy efficiency; hence, the process of restoring economic and social activities after the COVID-19 pandemic is in line with efforts to reduce GHG emissions.

Furthermore, Indonesia has also transitioned to renewable energy through the realization of energy policy and energy plan, as well as by increasing the percentage of the renewable energy mix from 11% in 2021 to 23% in 2025 and 31% in 2050. This paper also provides some information about the current energy situation. For instance, in 2021, coal energy was the largest supplier with a total energy of 558.782 million BOE and 180.509 million BOE for renewable energy. The equilibrium between energy availability, production, and consumption is interrelated. Its current conditions can be used to determine the next steps in the utilization of renewable energy. The sources available under conditions of energy balance in Indonesia from primary energy without biomass show the average growth by 3.5% yearly from 2015 to 2020. Meanwhile, energy production in 2020 shows a total of 443.1 million TOE, with 94.9% from fossil energy comprising oil, gas, and coal. Energy consumption per type in 2021 is currently experiencing an increase of 0.99%, which is 939.100 million BOE. Transportation is the sector with the largest energy consumption rate at 45.76%. This sector almost entirely uses gasoline as a fuel source. In accordance with the current energy consumption data, the use of fossil energy is still dominant and continues to increase. Therefore, with Indonesia's abundant renewable energy potentials, it needs to be utilized appropriately and maximally.

Optimizing renewable energy sources for power generation is part of the strategic plan to develop NZE power plants. Some of the performance targets for the renewable energy and energy conversion sub-sector in 2022 are the primary energy mix of 15.7%, with 366.4 MBOE produced. Several renewable power plants also support the government in reducing fossil fuels such as coal in steam power plants and these include:

a. Hydro energy sources have a potential of 95 GW, consisting of 75,000 MW and 19,370 MW of hydro and micro-hydro potential.

b. Geothermal energy sources are used in geothermal power plants with a potential of 23,965 GW. They are distributed on the islands of Sumatra, Java, Bali, Nusa Tenggara, Kalimantan, Sulawesi, Maluku, and Papua.

c. Bioenergy is used for several plants such as biomass, biogas, municipal waste, household, and power plants, etc. The total potential of bioenergy is 32,653.8 MW.

d. Solar energy has a potential of 207,898 MW at an average intensity of 4.80 kWh/m$^2$/day.

e. Wind energy, from the latest data in the third quarter of 2021; Indonesia has an installed capacity of 154.3 MW, while the target in 2025 is 255 MW.

Based on these conclusions, the present paper is expected to play a role in planning and implementing policies regulating the utilization of renewable energy. The electricity supply from fossil fuels will likely be used in the next few generations as a renewable energy reserve. The shift from fossil to renewable energy sources can contribute to the country's economy. The cost of this shifting needs to be also lowered by incorporating private sector investment in renewable energy projects.

**Author Contributions:** The authors confirm contribution to the paper as follows: study conception and design: N.A.P.; data collection: R.A.F., R.R., and M.S.S.; analysis and interpretation of results: N.A.P., S. (Suharno), and S. (Sukatiman); draft manuscript preparation: D.K.U. All authors have read and agreed to the published version of the manuscript.

**Funding:** This research was funded by Universitas Sebelas Maret, Indonesia (Grant Number 254/UN27.22/PT.01.03/2022); PUT-UNS; LPPM Universitas Sebelas Maret; Ministry of Education and Culture of Indonesia.

**Institutional Review Board Statement:** Not applicable.

**Informed Consent Statement:** Not applicable.

**Data Availability Statement:** Not applicable.

**Acknowledgments:** The authors would like to gratefully acknowledge Universitas Sebelas Maret, Indonesia (Grant Number 254/UN27.22/PT.01.03/2022); PUT-UNS; LPPM Universitas Sebelas Maret; Ministry of Education and Culture of Indonesia.

**Conflicts of Interest:** The authors declare no conflict of interest.

## Abbreviations

List of abbreviations

| | |
|---|---|
| AC | Alternating current |
| BaU | Business as usual |
| BOE | Barrel oil equivalent |
| BOPD | Barrels of oil per day |
| BOSS | Biomass operation system of Saguling |
| BSCFD | Billion standard cubic feet per day |
| CCS/CCUS | Carbon capture, utilization, and storage |
| CEDI | Clean energy demand initiative |
| CO$_2$ | Carbon dioxide |
| CO2e | Carbon dioxide equivalent |
| COD | Commercial operation date |
| COVID-19 | Corona Virus Disease 19 |
| GDP | Gross domestic product |
| GHG | Greenhouse gas |
| GW | Gigawatt |
| GWh | Gigawatt hours |
| Ha | Hectare |
| HEC-HMS | Hydrologic engineering Center-hydrology modelling system |

| HMS | Hydrology modelling system |
| kL | Kiloliter |
| LPG | Liquefied petroleum gas |
| LTSHE | Energy-saving solar light |
| $m^3$ | Cubic meter |
| MBOED | Million barrel oil equivalent |
| MEPS | Minimum performance standard |
| MMSTB | Million stock tank barrels |
| MW | Megawatt |
| MWh | Megawatt hours |
| MWp | Megawatt peak |
| NDC | Nationally determined contribution |
| NZE | Net-zero emission |
| PB | Sustainable development |
| Pb | Base load |
| Pp | Peak load |
| RK | Low Carbon |
| RON | Research octane number |
| SKKNI | Indonesian national work competency standards |
| SN | National standard |
| SR | Household connection |
| TOE | Ton of oil equivalent |
| USD | United States dollar |

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
