# Peer review of "Renewable Energy in Indonesia: Current Status, Potential, and Future Development"

_sustainability, doi:10.3390/su15032342_

Round 1

Reviewer 1 Report

The article deals with the past, present and future situation of renewable energy in Indonesia. I think it is important especially for island societies with similar climatic characteristics, as well as in the context of general international energy policy. I think the article needs some minor corrections. Below are some points that are worth considering.

1. Please consider transforming the current figures into more concise, less basic in the context of the fonts and styles used (currently these are default styles for Excel) - it will surely enrich both the substantive and visual side of the article. For example, the figures on page 4 could be placed side by side and take up less space, and figures 4 and 5 could be placed on the same coordinate system.

2. The manuscript lacks information about the databases used, keywords, as well as a quantitative list of publications on this topic in a multi-annual perspective. You can also draw up a diagram showing the process of creating an article, taking into account individual steps.

3. I would recommend changing the way of presenting certain content so that the presented data is shown as credibly as possible - e.g. the scale in Figure 9 does not start from 0 and the differences shown seem larger than in reality (similarly, e.g. in Figure 17). Please change this as wrong conclusions can be drawn.

4. The language and editorial side requires some corrections: e.g. % in Figure 10 and 11 is not included, it should be "households", not "house holds", "potential" or "capacity" instead of "potency" etc. Please, thoroughly verify text from this angle.

5. In order to systematize the layout of the text, I would recommend drawing up a diagram showing the layout of the content in the manuscript.

6. I miss the described subject in a broader perspective, for example in the context of energy policies other than national or in the context of international regulations concerning adaptation to climate change.

7. The format of the bibliographic list is incorrect. Please verify it with the journal's requirements. Where possible, DOI should be added for unique identification.

8. Please indicate the elements that distinguish this article from others. What practical application can it find? What is the gap in the current knowledge it is filling?

9. Please indicate the purpose of the article more clearly (currently it is one sentence).

10. Chapter 5.4.1 - the text mentions a potential map for the described solution. Maybe it would be worth actually quoting it in a graphic form in the article?

11. Please add a chapter / subchapter about the research area, i.e. Indonesia - mainly its geographical location, administrative division, and most of all the map of the country, because unfortunately the names of the provinces alone say little to people from outside this country.

12. General note: where possible, it is worth converting absolute values ​​into relative values ​​- e.g. Table 9 shows the technical potential of solar energy in 34 provinces of Indonesia, but it is difficult to say whether it is a large potential without information about the area of ​​the land or the number of inhabitants living in these provinces. It is worth converting this potential into km2 or per capita for each province separately.

I hope you will find my comments helpful. Good luck with revising your manuscript!

Author Response

We thank the reviewer for your comments and suggestions for improving the manuscript. We have corrected all comments to the best of our abilities and feedback is given in red text. Changes made to the script use yellow highlights. I submit the correction in the file attachment below.

Reviewer 2 Report

The manuscript, at the present form is not acceptable. Therefore,  in the following, I enlist some issues that need to be incorporated by the authors if they intend to resubmit.

·         There are too many grammatical errors like; “The main focus of this paper is provides a detailed analysis of the current status….”

·         The novelty and contribution of the paper are unclear.

·         The Introduction should make a compelling case for why the study is useful along with a clear statement of its novelty or originality by providing relevant information.

·         The results should be extended because there are too many figures / graphs, but proper explanation is missing.

·         Authors need to add more key findings in conclusion.

·         Add way forward or policies for its adaption.

Author Response

(The authors gave the same response as above.)

Reviewer 3 Report

The research paper is very interesting and may contribute into the literature. However, I have following comments for the improvement:

- Generally, the abstract of paper is based on research aim/purpose, research method, and key findings. Abstract of this paper is well written but it is required to highlight the key findings of the topic.

- First introduction needs to be expanded further to cover all the critical aspects of the topic.
I suggest author(s) to put more efforts and cite relevant articles. Following are some articles, which can be cited:

·         10.1007/s11356-022-22772-9

·         https://doi.org/10.1016/j.jclepro.2020.125624

·         https://doi.org/10.1016/j.jclepro.2021.129776

·         https://doi.org/10.1016/j.jclepro.2022.130501

·         https://doi.org/10.1016/j.jclepro.2020.120484

·         10.1016/j.techfore.2021.121417

- Research novelty and contribution should be highlighted in introduction section.

- A section detailing the literature review needs to be included. Similarly, describe in detail the study methodology used.

- Improve the writing of the document and align with the requirements of the journal.

- Results and discussion section: The paper presented and explained all the key findings but they did not discuss the findings with the help of previous published papers. Author(s) discussed their results very well but in a scientific paper, it is required to cover all the aspect and provide and cite the similar work of other researchers. I think author(s) need to polish this section, which will help to further highlight the researchers' work.

- Conclusion section is insufficient. Also, authors required to expand the policy implications according to their objectives and aim of the study.

General features

Further, author(s) should check the grammatical and English errors. I suggest author(s) to proof-editing to the entire manuscript, it will significantly help to improve English language.

Thank you again for giving me the chance to read this manuscript.

Author Response

(The authors gave the same response as above.)

Reviewer 4 Report

In my opinion the article I have been asked to review is neither a review article nor a research article under the given title. It presents statistics on the use of conventional and renewable energy sources in Indonesia. I consider to be unreasonable to include 21 figures and 11 tables, as the content presented in the paper is a duplication of what the graphics contain.  Moreover, the tables present statistics for different provinces in Indonesia, which means nothing to a potential reader outside the region. Incorrect captions are provided under the figures, e.g. Fig 7, Fig 9. The data presented in the figures do not cover compatible years. The editing of the paper itself, the graphics and the list of literature cited by the authors leave much to be desired. The conclusions, on the other hand, are a summary of the text and in places a repetition and duplication of the statistics presented in the earlier text

In my opinion, the article is not suitable for publication in the journal Sustainability.

Author Response

(The authors gave the same response as above.)

Round 2

Reviewer 2 Report

Authors have incorporated the comments, it can be published.

Author Response

We would like to thank the reviewers for their comments and suggestions for improving the manuscript. 

Reviewer 3 Report

Changes have been successfully addressed.

Author Response

(The authors gave the same response as above.)

Reviewer 4 Report

The article submitted for review has been developed and focused in some ways by the Authors. In the introduction, the purpose of the article was provided implicitly, and the structure of the paper was systematized by adding a Methods chapter, with an emphasis on the review nature of the presented content. In the further part of the paper, the Authors tried to develop the presented analyses in relation to their original version, which certainly improved the quality of this work in the perception. The list of literature was also improved. No less attention is still drawn to the presented data for different periods . Despite the inclusion of Figure 17 (no direct reference in the text), presenting renewable energy distribution in Indonesia, it is not readable. I stand by my opinion that the graphic part of the work is too elaborate, which is duplicated by the content included in the work. 

Author Response

We would like to thank the reviewers for their comments and suggestions for improving the manuscript. We have corrected all comments to the best of our abilities and our feedback is provided.

Round 3

Reviewer 4 Report

I believe that despite the effort put by the Authors of the article to improve the work, it does not present a very high scientific level. I stand by my comments in previous versions of the review.. However, I will honor the Publisher's decision.